# Diamond for High-Power, High-Frequency, and Terahertz Plasma Wave Electronics

**DOI:** 10.3390/nano14050460

**Published:** 2024-03-01

**Authors:** Muhammad Mahmudul Hasan, Chunlei Wang, Nezih Pala, Michael Shur

**Affiliations:** 1Electrical & Computer Engineering, Florida International University, Miami, FL 33174, USA; mhasa043@fiu.edu; 2Mechanical and Aerospace Engineering, University of Miami, Coral Gables, FL 33146, USA; wangc@miami.edu; 3Electrical, Computer and Systems Engineering, Rensselaer Polytechnic Institute, Troy, NY 12180, USA; shurm@rpi.edu

**Keywords:** diamond, terahertz (THz), electronics, single crystal growth, power electronics, high frequency FET, TeraFET

## Abstract

High thermal conductivity and a high breakdown field make diamond a promising candidate for high-power and high-temperature semiconductor devices. Diamond also has a higher radiation hardness than silicon. Recent studies show that diamond has exceptionally large electron and hole momentum relaxation times, facilitating compact THz and sub-THz plasmonic sources and detectors working at room temperature and elevated temperatures. The plasmonic resonance quality factor in diamond TeraFETs could be larger than unity for the 240–600 GHz atmospheric window, which could make them viable for 6G communications applications. This paper reviews the potential and challenges of diamond technology, showing that diamond might augment silicon for high-power and high-frequency compact devices with special advantages for extreme environments and high-frequency applications.

## 1. Introduction

Applications in renewable energy transport and distribution, power electronics, 6G communications [1], imaging [2], security [3], VLSI testing [4,5], IoT networking [6], sensing [7], and healthcare [8,9] have been the driving force behind the relentless quest for high-power and high-frequency electronic devices. In this context, diamonds with extraordinary properties have emerged as an extremely promising material. Among wide-bandgap materials, diamond has one of the largest bandgap energies of E_G_ = 5.46 eV, i.e., it is an ultra-wide bandgap semiconductor (UWBGS) with an impressive breakdown field of 10 MV/cm and an exceptional thermal conductivity of 23 W/cmK [10]. These attributes position diamond as a compelling candidate for applications demanding high-power, high-frequency, and the ability to withstand extreme temperatures or operate in a radiation environment.

However, diamond’s constraints related to material growth and especially doping created obstacles in the production of high-quality diamond devices. While substantial strides have been made in surmounting these obstacles [11,12,13], issues related to substrate size, lattice quality, and n-type doping persist, necessitating sustained research endeavors. Within the domain of radio frequency (RF) devices, state-of-the-art diamond devices have shown impressive performance metrics—cutoff and maximum frequency, as well as output power density—surpassing those of devices implemented in many other material systems [14,15,16]. Nonetheless, diamond RF devices have not yet eclipsed the performance benchmarks set by AlGaN/GaN high-electron-mobility transistors (HEMTs) [17]. Diamond devices still lack in terms of using the full potential of diamond material properties.

Diamond-based electronics hold promise of operating in the sub-terahertz (sub-THz) and terahertz (THz) frequency regimes. By capitalizing on plasma waves within Terahertz Field-Effect Transistors (TeraFETs), diamond offers an innovative way that avoids traditional electronics paradigms [18,19,20]. This new development holds the promise of achieving room temperature operation in the millimeter-wave (mm-Wave) frequencies, heralding advanced THz applications, including the realization of 300 GHz (6G) communication systems [20].

This paper presents an overview of the state-of-the-art in diamond-based high-power and high-frequency devices. It also reviews the material preparation for the devices and underscores the potential of diamond devices capable of operating in the sub-THz and THz regions. Simultaneously, it also highlights the challenges of fully realizing the potential of diamond electronics.

## 2. Electronic Material Properties of Diamond

Diamond is a unique material with exceptional electronic properties. It has an ultra-wide bandgap, high thermal conductivity, and the largest carrier mobility among wideband gap materials [21]. At room temperature, diamond is a good electrical insulator, but it acts as a semiconductor material when doped with impurities such as boron, nitrogen, or phosphorus. These special set of qualities make diamond desirable in high-power, high-frequency electronic applications. In this section, we will discuss the diamond’s key electrical characteristics and contrast them with those of competing materials.

### 2.1. Comparison Figures of Merit

Diamond surpasses other materials by a wide margin in many metrics, including Baliga (power handling capabilities) [22], Johnson (RF power capability) [23], and the combined factor of merit [24] (see Figure 1). The combined figure of merit (CFOM) is given by:(1)$$CFOM=\frac{\chi E_{B}^{2}K\mu v_{s}}{{\left(\chi E_{B}^{2}K\mu v_{s}\right)}_{silicon}}$$
where χ is thermal conductivity (W/cm degree K), $E_{B}$ is breakdown field (kV/cm), μ is low field mobility (cm^2^/Vs), $v_{s}$ is saturation velocity (m/s), and K is the dielectric constant. Table 1 lists the material parameters used to calculate the figures of merit.

We also included the comparison of other figures of merit (FOM), including Baliga’s high-frequency FOM (BHFOM) [22], Keye’s FOM (KFOM) [25], Huang’s temperature FOM (HTFOM) [26], Huang’s chip-area FOM (HCAFOM) [26], and Huang material FOM (HMFOM) [26]. BFOM rates the power handling capability of the material. On the other hand, BHFOM and JFOM can rate the high-frequency and power operations. KFOM provides information about the thermal limitations of switching devices. Table 1 compares the material properties along with different FOMs among diamond and other materials. It shows that diamond surpasses other WBG materials in every category due to its excellent material properties.

**Table 1 nanomaterials-14-00460-t001:** Comparison of material properties and different FOMs for diamond and other WBG materials [24,27,28,29,30,31].

Material Properties	GaAs	Si	4H-SiC	GaN	BN	AlN	Ga_2_O_3_	Diamond	Comment
Energy bandgap (eV)	1.42	1.1	3.2	3.44	6.4	6.2	4.8	5.47	High-temperature operation
Dielectric constant	12.9	11.8	9.7	9	7.1	9.76	10	5.7	-
Breakdown field (MV cm^−1^)	0.4	0.3	3	5	17.5	15.4	8	10	High voltage applications
Effective mass	0.067	0.19		0.24	0.43	0.4	0.297	0.74 (hole) 0.36 (electron)	Long momentum relaxation time
Thermal conductivity (Wm^−1^ K^−1^)	55	150	500	130	2145	1000	27	2200	High-power applications
Electron saturation velocity (cm s^−1^)	1.1 × 10^7^	0.86 × 10^7^	3 × 10^7^	2.5 × 10^7^	4.3 × 10^7^	1.3 × 10^7^	2 × 10^7^	2.7 × 10^7^	
Hole saturation velocity (cm s^−1^)	-	0.8 × 10^7^	1.2 × 10^7^	-	-	-	-	1.2 × 10^7^	High-frequency operation
Electron mobility (cm^2^ V^−1^ s^−1^)	8500	1450	900	440	825	426	300	7300	
Hole mobility (cm^2^ V^−1^ s^−1^)	-	480	120	200	500	-	-	5300	
BFOM	15	1	554	188	-	35,497	3214	23,068	Power handling capability
CFOM	5	1	594	162	-	6518	53	124,424	
BHFOM	-	1	58	237	-	-	158	12,510	Takes switching losses into account
JFOM (10^23^ ΩW s^−2^)	1.36	2.3	900	490	-	64	1236	2530	RF power capacity
KFOM (10^7^ WK^−1^ s^−1^)	-	10	53	17	-	-	2	218	Thermal limit of transistor characteristics
HMFOM	-	2	-	18	-	-	22	72	
HCAFOM	-	1	58	192	-	-	280	3887	Reduction of chip area

### 2.2. Carrier Mobility

The carrier mobility of diamond is among the highest of any known material. Intrinsic carrier mobility, which is the mobility of carriers in pure diamond without the presence of impurities or defects, has been measured to be around 4500 cm^2^/Vs for electrons and 3800 cm^2^/Vs for holes at room temperature for single crystal CVD grown diamond [32]. Higher values of electron (7300 cm^2^/Vs) and hole (5300 cm^2^/Vs) mobility were measured in ultrapure diamond crystal using the time-resolved cyclotron resonance (TRCR) method [21]. For comparison, 4H-SiC showed mobility for electrons and holes at 900 cm^2^/Vs and 120 cm^2^/Vs, respectively [33,34]. Diamond’s distinctive crystal structure, which is made up of a three-dimensional network of sp^3^ hybridized carbon atoms, is responsible for its remarkable carrier mobility. Strong covalent connections between the carbon atoms in this structure result in a high degree of stiffness and a high-frequency of lattice vibrations, which decrease carrier scattering [35]. Diamond hole mobility decreases at temperatures above room temperature (300–400 K), which can be attributed to the scattering by acoustic phonons [32] and incomplete ionization of impurities [36]. Studies show that the hole mobility comes down to approximately 1000 cm^2^/Vs at 500 K from approximately 2000 cm^2^/Vs at 400 K [29]. For the devices where doping is needed to make the p-type and n-type materials, these large mobilities cannot be sustained because the dopant scattering decreases mobility. A boron (B)-doped CVD diamond showed hole mobility of 1500 cm^2^/Vs, 1000 cm^2^/Vs [37], and 450 cm^2^/Vs [29] for B concentrations of 5 × 10^16^ cm^−3^, 2 × 10^18^ cm^−3^, and 1 × 10^19^ cm^−3^, respectively. In a phosphorous-doped CVD grown diamond homoepitaxial film, the Hall mobility was 660 cm^2^/Vs at room temperature with a phosphorous concentration of 7 × 10^16^ cm^−3^ [38]. The hole mobility during surface transfer doping was as high as 680 cm^2^/Vs for a sheet hole carrier concentration of 5 × 10^12^ cm^−2^ [39,40]. Recently, a new heterojunction (h-BN/diamond) method has been introduced in the surface transfer doping, resulting in reducing the surface charged impurities [41]. This methodology demonstrated a hole carrier mobility of 300 cm^2^/Vs for a 5 × 10^12^ cm^−2^ sheet hole density [42]. However, even these small values of carrier mobility of diamond with doping are way higher when compared with other competing wide band gap materials such as p-GaN (30 cm^2^/Vs) and p-SiC (<17 cm^2^/Vs) [40]. Figure 2 shows the relative contributions of different scattering mechanisms to the mobility of p-type diamonds [43].

### 2.3. Thermal Properties

Due to the strong covalent connections between the carbon atoms organized in a crystalline lattice structure, diamond has remarkable thermal characteristics, having high thermal conductivity, a low thermal expansion coefficient, and an extremely high melting point. The thermal conductivity of diamond ranges from 700 W/mK at 773 K to 2300 W/mK at room temperature [40,44]. Hence, it is a superior heat conductor to most materials, making it an ideal candidate for heat sinks and high-power electronic devices. Moreover, diamond has a very low coefficient of thermal expansion of 1 × 10^−6^/K (at 300 K) compared to other wide bandgap semiconductors (AlN = 5.3 × 10^−6^/K, GaN = 3 × 10^−6^/K, SiC = 2.8 × 10^−6^/K) [45,46,47,48]. The 3550 °C melting point of diamond is also a significant characteristic for high-temperature and harsh environmental applications [44].

### 2.4. Saturation Velocity of Carriers

The saturation velocity is an important parameter for understanding the electronic properties of diamond and its potential for use in high-speed electronic devices. Usually, the electron drift velocity saturates in high electric fields due to the emission of optical phonons by energetic electrons, and the electron saturation drift velocity is an increasing function of optical phonon energy [40,49]. Diamond has a very high optical phonon energy of E_opt_ = 160 meV [50], which results in fairly high saturation velocity. The saturation velocity of diamond has been measured experimentally and found to vary from 0.85 to 1.2 × 10^7^ cm s^−1^ and 1.5 to 2.7 × 10^7^ cm s^−1^ for holes and electrons, respectively [29,44,51]. This translates into a higher upper bound of the diamond cutoff frequency (fT = vs/(2πL)). These values are similar to the saturation velocities of other common semiconductors such as SiC and GaAs. However, diamond can reach this velocity at a lower electric field of approximately 10 kV cm^−1^ compared to SiC and GaN (see Figure 3) [29].

### 2.5. Breakdown Field

For electronic devices where fast switching is required, a smaller device size is an advantage, and a high breakdown field allows a smaller device size [29]. Diamond has an exceptionally high breakdown field, estimated to be around 10 MV/cm [55]. In comparison, the breakdown field strength of other materials such as 4H-SiC and GaN is much lower, at around 3 MV/cm and 5 MV/cm, respectively [29]. Comparing the breakdown voltage of semiconductor devices constructed from various materials to the specific ON resistance is an effective method of comparison. Figure 4a illustrates the relationship between ON resistance and the breakdown voltages for different materials. Also, Figure 4b shows the breakdown field with respect to the different energy band gaps of various semiconductor materials.

### 2.6. Carrier Lifetime

The carrier lifetime of a single-crystal, high-quality CVD diamond ranges from nanoseconds to several microseconds [32], comparable to SiC [58,59]. Nitrogen impurities in diamond can serve as electron recombination hotspots, greatly reducing the carrier lifetime. The carrier lifetime in a highly doped diamond can be as low as nanoseconds or picoseconds [60,61].

## 3. Material Quality and Growth Techniques

### 3.1. Substrate Quality

Diamond substrates of high crystalline quality and high purity are needed for various applications. Usually, there are N_2_ contents in the diamond substrate [62]. Based on the impurity concentrations available, synthetic and natural diamond substrates are primarily classified into two main types: Type I and II. Type I contains a higher percentage of N_2_ impurity (from less than 1 ppm up to several thousand ppm), which can be detected using IR spectroscopy [28]. However, high-quality Type I diamonds used in electronic and optical applications typically have a nitrogen concentration of less than 100 ppm [63]. Diamond, where these nitrogen atoms can be aggregated by replacing carbon atoms is classified as Type Ia. On Type Ib diamond substrates, nitrogen impurities can also be isolated from each other. Although its crystalline quality is worse, Type Ib is often used for electronics applications because of its low cost and reasonable level of dislocation density (10^5^ cm^−2^). The aggregated nitrogen atoms in Type Ia substrates can come in two forms: Type IaA, where the nitrogen atoms are in pairs, and Type IaB, where a vacancy is surrounded by 4 N atoms [28].

When the impurity level of N_2_ in a diamond substrate is too low (less than 10^17^ cm^−3^) to be detected in IR spectroscopy, diamond is categorized as Type II. Type II is also sub-categorized into Type IIa, the purest of all diamond substrates (produced in a high-pressure, high-temperature (HPHT) process), and Type IIb, with the boron concentration being higher than N_2_, making it the p-type diamond substrate [28]. Nitrogen and boron impurity concentrations in Type IIa are the lowest, and this material is the purest and most costly. The classification of diamond substrates is graphically represented in Figure 5.

### 3.2. Diamond Growth Technologies

Diamond is synthesized mainly using two different growth technologies: high-pressure, high-temperature (HPHT) and chemical vapor deposition (CVD). The HPHT process is mostly used in industries to produce synthetic diamond [63] and highly pure and crystalline (<10^3^ cm^−3^ dislocation density) Type IIa diamond substrates. However, the substrate size is limited to below 1 cm^2^ [28]. The HPHT process uses the high-pressure, high-temperature diamond area of the carbon phase diagram, where diamond is very stable. On the other hand, CVD uses the graphite region of the carbon phase diagram [63]. Figure 6 shows the conditions for the HPHT and CVD diamond synthesis processes in the carbon phase diagram [64]. The CVD process is favored for electronics applications because it can produce a large substrate while keeping the dislocation density in an acceptable range.

#### 3.2.1. Diamond Growth by HPHT

Solubility gradient, temperature gradient, no-catalyst conversion, and shock compression are the four major types of HPHT diamond synthesis techniques [3]. Ni, Co, and Fe-like metals are used in molten form in the solubility gradient method to solve the diamond source (graphite) [4]. High-temperature, high-pressure are very stable conditions for diamonds. Most high-quality diamonds for industry applications are produced in this way. However, the substrate size is limited to less than 1 mm^2^ [65,66,67,68,69], which is why HPHT is not suitable for larger size wafer technology.

In the temperature-gradient approach, a diamond seed crystal is used to support nucleation on the downside of the HPHT container. The diamond source is kept on the top side of the container, and in between there is a solution of metals such as Ni, Co, Fe, Mn, Cr, Ta, or Nb. The condition set for the diamond to grow is 5–6 GPa pressure and 1300–1600 °C temperature [63,64]. Inside the vessel, a temperature gradient of 20–50 °C is created where the top side/diamond source is on the hotter side than the seed crystal/bottom side. This melts the diamond source in the Fe, Ni, Co, Mn, Cr, Ta, or Nb solution, making it supersaturated. The melted diamond source is then transported to the seed crystal as HPHT crystallized diamond [64]. Depending on the different growth rates at different crystal directions, HPHT can produce diamond crystals shaped in various ways, such as cubes or octahedral [70,71,72,73,74,75,76,77]. To produce large diamond crystals, the ‘no seed crystal’ method is used, and sometimes it results in incorporating nitrogen impurities in the diamond. These can be detected by IR testing. Highly pure carbon source, and Ti is used to get rid of the nitrogen impurities from the metal solution [78]. In the process, TiC is produced as a byproduct, which is later removed by adding Ag and Cu [63].

#### 3.2.2. Diamond Growth by CVD

As mentioned above, the stable graphite region in the carbon phase diagram is used for CVD diamond growth. Unlike HPHT and natural diamond synthesis processes, CVD processes are more complex. The earliest work on producing synthetic diamonds in the CVD process dates from the 1980s [79]. The process involves H_2_ and CH_4_ as the reactant gases where, methane is used as the source for the diamond [80]. In the process called chemical vapor deposition, these gases are involved in a reaction in the gas-phase and are deposited on the substrate to grow epitaxial diamond. The main stages and parameters of the CVD process are shown in Figure 7 [80,81]. Among the major elements of the process, the reactant gas is the most important. Next comes the energy for the reactions to take place in the plasma condition, which can be supplied using different sources. Depending on the energy source type, the whole structure and the parameters of the CVD process change [82]. Lastly, to facilitate the efficient output from the reactions of the gases, the correct temperature, pressure, flow rate, density distribution, and other parameters are maintained.

One of the most critical points in CVD diamond growth is the presence of a radical H_2_ (H^*^) atom. The hydrogen atoms could split neutral hydrocarbons into reactive radicals such as CH_2_. This excited hydrocarbon may create graphite (trigonal sp^2^) or diamond (tetrahedral sp^3^)-linked carbon on this exposed carbon [83]. However, this sp^2^ (graphite) bond is more stable than the diamond bond. The radical hydrogen inhibits the development of this graphite bond and facilitates diamond growth [84]. Also, it halts the growth of the polymer of the hydrocarbon, which might hinder diamond growth.

Since 1993, when the main chemical process for the formation of diamond from gaseous hydrogen and hydrocarbon compounds was first proposed [80], diamond CVD growth techniques have been constantly improved. Single-crystal epitaxial growth of diamond can be carried out using the CVD process. Two different methods are currently being studied to produce high-quality single-crystal diamonds. In one method, a diamond substrate is used to grow the epitaxy, which is called homoepitaxy. In contrast to that, a different material substrate other than diamond is used in heteroepitaxy. To create plasma and to get the activation energy for the reactant gases different plasma sources are used. They are classified as microwave [85], laser induced [86], radiofrequency [87], direct current [88], hot filament [89], and chemical activation (see Figure 7). Among them, microwave plasma-assisted plasma CVD (MPCVD) is a well-studied [90,91,92,93,94,95] and established diamond CVD growth technology. It can produce high-quality and highly pure (nitrogen concentration < 5 ppb) diamond crystals. These diamond substrates can be used in electronics applications [11]. The diamond substrates grown this way have excellent electrical (two orders higher electron mobility than natural diamond of type IIa [32]) and thermal (approximately 24 W cm^−1^ K^−1^ at room temperature and 278 W cm^−1^ K^−1^ at 63 K [96]) properties. However, improving the growth rate, minimizing defect densities, and achieving longer continuous growth are required to achieve CVD diamond growth on an industrial scale.

Several process factors influence CVD diamond synthesis, including reactor gas pressure [97,98], concentration of diamond source (methane) [99], absorbed microwave power density [97,100], temperature [97,100], nitrogen impurity [97,101], substrate holder shape [102], and others. Growth rates up to 165 µm h^−1^ were reported for 18 mm thick single crystal diamond (SCD) development where the gas pressure was 350 Torr. For this process, a recipe was developed by integrating multiple advantageous characteristics of the process parameters [103]. It was a hard and lengthy process of separating the effects of each process parameter to optimize the reactor condition and enhance the growth rate. A faster method has been reported where multiple parameters were assessed in a single cycle, which cut the test time significantly. From linear growth of 5–10 min, the growth rate was determined with great precision [100,104]. The relationship between the concentrations of methane and growth rate at different temperatures was identified. Typically, a higher concentration of methane increases the growth rate [104], but it damages the crystallinity of the diamond. The growth rate of typical high-crystalline diamond substrates is <1 µm h^−1^ [105,106,107,108]. Typically, the methane gas flow is pretty low in contrast to the total source gas flow (relative ratio < 1%) [107]. Nonetheless, higher pressure [109] and higher microwave [110,111] power systems have been reported as a solution to this issue, even with a low methane concentration (4%). The increment in pressure from 100 Torr to 600 Torr results in greater microwave power (approximately 1800 W cm^−3^) absorption. This produced an excellent growth rate of approximately 60 µm h^−1^ [109]. A growth rate of over 10 µm h^−1^ was recorded when deposition was conducted with 3.8 kW microwave power in the MPCVD process [110]. The size limitation of homoepitaxial diamond growth is another important issue. Recent studies showed that by growing a thick layer, the initial seed size can be increased by using the lateral growth of diamond [112,113]. Moreover, half inch diamond substrates were successfully produced using side face growth of diamond and lift off processes [114]. Another promising way of increasing the homoepitaxial diamond substrate size is the mosaic growth method [115,116]. A single crystal diamond is grown first in similar small seeds, which are then congregated together to make one mosaic array. This mosaic array is then used to grow the CVD layer, resulting in a larger substrate size [117,118]. Usually, these mosaic wafers are about 40 × 60 mm^2^ in size and made from 10 × 10 mm^2^ small substrate unit cells [118]. However, the joint areas of the mosaic substrates suffer from high defect densities and strain [119,120]. A tungsten impurity enriched diamond buffer layer along the junction boundaries blocked the defect transfer to the epilayer [120,121].

Substrate selection is a crucial factor in the heteroepitaxy CVD diamond growth process. The right substrate should have a combination of different important parameters, such as an appropriate lattice constant, the stability of the substrate under plasma conditions, and many others [11]. A number of materials, such as Si [122], SiC [123], Ir [124], Co [125], c-BN [126], Re [127], Al_2_O_3_ [128], Pt [129], Ni [130], and TiC [131], have been explored for diamond heteroepitaxy growth. Iridium was used as a buffer layer on top of the metal oxide layers on Si substrates to reduce the cost and produce larger substrates [132,133]. The concept of a multilayer substrate was an excellent addition to the technology. This allowed us to grow iridium on commercially available large substrates of Si (12 inch), MgO (2 inch), and sapphire (8 inch) and produce large diamond substrates [11]. MgO [134] and SrTiO_3_/Si [135] wafers were used to successfully produce 10 × 10 mm^2^ and 7 × 7 mm^2^ heteroepitaxial diamond films, respectively. Recently, studies on the synthesis of a 3.5 inch diamond substrate on Ir/YSZ/Si [136] and a 2 inch substrate utilizing Ir heteroepitaxy have been reported [137,138]. Though heteroepitaxial diamond has the advantage of being larger in size, it still suffers from a higher dislocation density (10^8^–10^9^ cm^−2^) than homoepitaxial diamond (10^2^–10^5^ cm^−2^). The dislocation density tends to decrease with thicker films [139]. The performance of the devices built with these diamond materials heavily depends on their crystalline quality. Many applications require larger wafer sizes. According to several recent studies, while patterning diamond nucleation, the stripes’ direction affects how dislocations are transferred to the developing film [140,141,142]. Using these discoveries, multiple research groups were able to reduce the dislocation density to below 9 × 10^6^ cm^−2^ [143,144]. A dislocation density of 1.4 × 10^7^ cm^−2^ was reported for a 1 inch diamond substrate grown on sapphire with excellent crystal characteristics [137]. Current advancements and developments in technology have incorporated this material into various applications. Heteroepitaxial diamond was used to make p-i-n [145], Schottky barrier diodes [146], and a high-voltage field-effective transistor [147].

### 3.3. Doping

As we discussed in the previous section, diamond has tremendous potential for making high-power and high-frequency devices. Both p- and n-type diamonds of good quality are required in order to achieve the full advantage of their excellent material properties [11,148]. Unlike silicon, most of the novel wide band gap semiconductors do not have donors and acceptors with low activation energies [28].

#### 3.3.1. P-Type Doping

Boron and aluminum are typical p-type dopants for diamond [149]. B is a natural impurity in diamond [64,150] and is the most established and developed p-type dopant [148,151,152]. Boron and carbon have 0.088 nm and 0.077 nm covenant bond radii, which are close enough to produce good quality boron-doped p-type diamond [153]. Boron’s activation energy is 0.37 eV [154]. There are multiple ways to produce boron-doped diamonds, such as CVD growth through gas phase manipulation [151], ion-implantation [155], and high-temperature diffusion [156]. In ion-implantation, highly energized dopant ions are bombarded into a diamond layer. Diamond is an ultra-hard material with sp^3^ hybridization, and it is hard to break those bonds to replace carbon atoms with dopants. Once the bonds are broken, the diamond tends to go back to more stabilized sp^2^ graphite formation, where defects form [149]. Different methods, such as annealing and etching, were developed to overcome this issue [155,157,158]. Ion-implantation with annealing and etching can achieve outstanding ohmic contact for p-type diamonds [158]. However, ion-implantation cannot provide a uniform distribution of dopants (it produces Gaussian distributions for a fixed energy of impinging ions [159]). In order to get at least close to a uniform distribution, multiple ion energies are needed. Moreover, because of the hardness of diamond, the ion implantation method leaves many defects in the material and makes it harder to get a good quality crystal [160,161]. More recently, the in situ method (doping during diamond growth) [151] has become more popular due to the complexities of the ex-situ (doped after growing diamond) [155,156] methods. Doping boron during the CVD growth of diamond is the most established method for producing p-type diamond. The source gases of boron are incorporated with the carbon source gases in the CVD process. Diborane [162], trimethylboron, triethylboron, boron oxide dissolved in alcohol [163], BCl_3_ [164], etc. are used as the sources for B. Studies showed that boron incorporation prefers the (111) diamond growth plane. [165]. The crystallinity and surface structure of diamond greatly impact the carbon replacement by boron in the diamond lattice [166,167]. Although the activation energy tends to reduce with high dopant concentration [168,169], it also increases stress [170] and dopant proximity effects [171,172] in the diamond structure, creating defects. This increase in defects results in lower mobility for carriers. Different studies recorded mobility of 120 cm^2^/Vs for a delta-doped thin layer (1 nm) with a boron concentration of approximately 4 × 10^20^ cm^−3^ [173,174]. Heteroepitaxial diamond substrates were explored for B-doping as well. Recently, different groups have investigated Ir [175] and other heteroepitaxial diamond substrates [176,177] to grow B-doped epilayers and overlayers (p+ and p−) of different thicknesses with B concentrations ranging from (10^15^–10^20^ cm^−3^). They have also reported on the Hall mobility of 390 cm^2^/Vs for a B concentration of 1 × 10^18^ cm^−3^ in a diamond epilayer [176]. However, boron-doped p-type diamonds have demonstrated Hall mobility as high as 2200 cm^2^/Vs at room temperature [178]. Some of the recent studies have reported on converting diamond to an insulator, metal, or superconductor by tuning the B concentration [168,169,179]. The production of p-type diamond is advanced and useful enough to make semiconductor devices such as field-effect transistors and Schottky diodes.

#### 3.3.2. N-Type Doping

Producing n-type diamonds with good carrier mobility and conductivity is problematic. Gathering suitable donor atoms with compatible growth technology is the research focus in the diamond electronics sector. Numerous potential elements were investigated as donor atoms; however, none of them produced satisfactory results [148] due to deep donor activation energies and poor doping efficiency. N, P, O, and S are on the frontlines of the n-type dopants for diamond, with donor activation energy levels of 1.7, 0.6, 0.32, and 1.5 eV, respectively [180]. Among them, phosphorous has shown promising results so far with its acceptable donor activation energy level [181,182]. The covalent bond radius of P is 0.117 nm, which is a big mismatch with carbon (0.77 nm). This makes it tough to incorporate P into diamond [28]. However, studies showed that the (111) diamond plane is a good match for P-doped diamonds to grow. The Hall mobility of 28 cm^2^/Vs with an activation energy of 0.55 eV was achieved. [183]. Nonetheless, P-doped diamond was also synthesized on a (100) oriented diamond surface [184], but much later than the synthesis of P-doped (111) diamonds [182]. After the first successful synthesis of P-doped diamonds with a Hall mobility of 23 cm^2^/Vs, many studies reported high quality P-doped diamonds [185,186,187]. Usually, PH_3_ is used as the source for P [182], but P_2_O_5_ [188] and liquid tertiarybutylphosphine (TBP) [189] were also utilized as sources for P, which resulted in a Hall mobility of 50 cm^2^/Vs (at a carrier concentration of 10^15^ cm^−3^), and 110 cm^2^/Vs, respectively. Until now, the highest mobilities were recorded for n-type diamonds with lightly doped phosphorous: [P] = 7 × 10^16^ cm^−3^ (carrier concentrations of 10^11^ cm^−3^) and [P] = 2 × 10^15^ cm^−3^ (carrier concentrations of 10^10^ cm^−3^) are 660 cm^2^/Vs [38] and 1060 cm^2^/Vs [186], respectively at room temperature. Moreover, a high concentration of P (10^20^ cm^−3^) can be used to create ohmic contact with diamond, which is essential for semiconductor devices [186]. Among other single element donors, N was used for doping, but it has a very deep donor energy level (1.7–2 eV), which results in very few ionized electrons [180]. More recent reports show improvement from this condition with better conductivity (approximately 140–150 Ω^−1^ cm^−1^), carrier concentration (10^20^ cm^−3^), and mobility (1–1.5 cm^2^/Vs) for N-doped diamond [190,191,192]. These results were achieved in ultra nanocrystalline diamond films, where grain boundaries are responsible for the improved conductivity [193,194]. This shows that N-doped diamonds are still far behind P-doped diamond technology. Though O has the lowest activation energy, it results in low mobility [195,196]. Theoretical studies show that the bond between C and O is stronger than C and C, implying that O could be a potential perfect donor impurity for diamond [197]. O-doped diamond was successfully synthesized by the implantation process [195,196,198]. However, studies showed that high oxygen can slow the growth of diamonds and increase their defect density [199]. Moreover, a simple substitutional technique for O will result in deep donor states and a smaller carrier concentration [200]. The status of the S-doped diamond technology is at a similar stage to that of O, as both underlying mechanisms are still being discussed. S-doped CVD grown diamond was successfully synthesized, with H_2_S being the source for S [201,202]. However, there were unintentional B atoms in the diamond, creating acceptor levels with an activation energy of 0.37 eV, whereas S was responsible for the 1.55 donor activation energy level, which is as deep as N [201]. Furthermore, theoretical investigations indicate that S has a much deeper donor level than P and has a lower bulk solubility than P [203]. Some other single elements dopants such as Ti [204], halogens (F, Cl, Br, and I) [205], As [206], Be [207], and Mg [207], are being investigated, but they are still in the exploratory phase. As there is no single-element dopants for diamond that produces optimum results, a novel co-doping approach employing multiple elements is being investigated. Incorporating B can make S a preferred shallow donor dopant for diamonds by improving the solubility and conductivity of S [208]. Additionally, B-P co-doped diamonds ere realized using ion-implantation, with a significant improvement in conductivity and Hall mobility [209]. Co-doping can also improve the lattice quality of p-type diamonds. The performance of B-doped p-type diamonds was enhanced in every metric, including conductivity, carrier concentration, and lattice structure, by using B-H co-doping [210]. It was also reported that double-acceptor-donor (B-B-As and B-B-P) impurities improve the lattice structure of p-type diamond [211].

#### 3.3.3. Surface Transfer Doping

An alternate way of creating the p-type conductance in diamonds is through the surface transfer doping method. Unlike other material systems, surface/interface roughness has a positive impact on the diamond hole mobility [212]. A diamond surface with hydrogen termination displays p-type surface conductivity and negative electron affinity after being exposed to air [213,214,215]. By using hydrogen plasma or annealing at 500 °C in an environment of hydrogen, it is possible to create H-terminated diamond surfaces [216,217]. Due to its reliance on the diamond’s surface’s hydrogen termination, this mechanism is strongly tied to the diamond’s surface. An illustration of the process is shown in Figure 8.

The formation of 2DHG in the vicinity of the diamond surface can be attributed to the added H atoms, which act as acceptor impurities, and the C-H dangling bonds, which cause upward band bending [214,219]. Currently, a complete explanation of surface transfer doping cannot be given by a single model [218]. Nevertheless, in the last ten years alone, rapid advancements have been made in this area. A detailed review of diamond surface transfer doping is given in [218]. To better understand the mechanism of surface transfer doping and to improve the performance of surface conductivity, various electron acceptor materials, including molecular species such as fullerene (C_60_) [220], fullerene variants (C_60_F_18_, C_60_F_36_, and C_60_F_48_) [221], tetracyanoquinodimethane (F4-TCNQ) [222], NO_2_ [223,224], O_3_ [225], and metal oxides such as molybdenum trioxide (MoO_3_) [226], vanadium pentoxide (V_2_O_5_) [227,228], tungsten trioxide (WO_3_) [227,229], rhenium trioxide (ReO_3_) [229], niobium pentoxide (Nb_2_O_5_) [227], chromium trioxide (CrO_3_) [230], and Al_2_O_3_ [231,232,233,234], were studied. They demonstrated mobilities ranging from approximately 10 to 120 cm^2^/Vs with carrier densities ranging between 10^12^ and 10^14^ cm^−2^ [218]. With carrier densities above 1 × 10^14^ cm^−2^, metal oxides (MoO_3_, V_2_O_5_, and WO_3_) currently have some of the lowest documented sheet resistances [226,227,229]. However, the lowest reported sheet resistance (719 Ω/□) of H-terminated diamond surfaces till now was attained using NO_2_ as the electron acceptor medium, where the hole density was up to 1.5 × 10^14^ cm^−2^ [223].

## 4. Diamond-Based High-Power Electronic Devices

Diamond electronic devices are projected to have a huge impact in the high-power electronic device area. Both unipolar and bipolar diamond devices have been proposed and developed.

### 4.1. Schottky, Diodes, and Bipolar Junction Transistors (BJTs)

Lack of n-type dopants with low activation energy in diamonds stimulates interest in p-type unipolar devices such as Schottky barrier diodes (SBDs) [235,236]. The advantages are low resistance [237], high doping concentrations [238], and high carrier mobility [239]. So far, the highest reported breakdown voltage for SBD is over 10 kV working at high-temperatures [240]. For high-power devices, a good contact is a critical issue as the electric field increases immensely around the sharp and rough contacts, resulting in early breakdown. The Schottky barrier height (SBH) in these diodes is stable in the range of 1.2–3.4 eV [241,242] depending on the Schottky metal and the surface termination process. Studies found that oxygen surface termination provides stable performance for the device [225,243,244]. Different Schottky metals (W, Mo, Cu, Pt, Ru, and Zr) have been explored for minimizing the leakage current and achieving high voltage operation [245,246,247,248,249]. Recently, p+ Si/p-diamond heterostructure diodes have been reported where Surface Activated Bonding (SAB) was used to form the heterojunction [250]. This method achieved a reduction in the turn on voltage and reverse bias current with a high rectification factor and a breakdown voltage of 200 V. Different diamond high-power diodes have been developed using Schottky junctions, such as vertical SBD [251], pseudo-vertical SBD [247], and Schottky PN diode (SPND) [252]. Typically, the vertical SBDs show breakdown when the voltage is in the range of 1.8–3.7 kV (the average electric field is 7.7 MV/cm) [236,253,254,255]. However, there is a scarcity of large single crystal diamonds for vertical SBDs, and the breakdown voltage may be explained by the crystal defects. A pseudo-vertical structure can solve this problem where the highly doped p-type layer is grown at high-pressure and high-temperature on a diamond substrate and the ohmic contacts are grown on the p+ layer. It boosts the current density [256]. Figure 9 illustrates schematics of different diamond junction power devices.

The issue with the bipolar devices is that their stored-up charges in the p-n junctions result in high voltage drops during the ON state. Also, it takes a longer time to deplete the charges during the reverse bias, inhibiting the high-frequency applications. When the p-n junction was created for BJT development using boron and nitrogen-doped diamond, the high resistivity of the nitrogen-doped base layer limited the device’s operation [257,258]. Using a high concentration of phosphorous dopants (shallow activation energy level than nitrogen) (10^22^ cm^−3^) improved the properties of the n-type layer [12]. Diamond orientation <100> was used to grow this n-type layer [13,259]. Several reports described the development of high quality p-n [260,261] and p-i-n [262] diamond junctions. The p-i-n structure has an intrinsic layer sandwiched between the p and the n layers, which supports a high reverse blocking voltage. The highest breakdown voltage of 11.5 kV was achieved using p-i-n structure [262]. However, these devices have a high ON resistance compared to unipolar Schottky p-n (SPND) [252] and Schottky p-i-n structures [263]. Current density of 60 kA/cm^2^ was reported to be achieved by SPND, where the ON resistance was 0.03 mΩ.cm^2^ [252].

Significant progress has been made in n-type doping techniques, enabling the fabrication of bipolar junction transistors (BJTs) [264,265,266]. The introduction of the n+ layer has played a crucial role in addressing the issue of the high resistivity of the n-type base layer. It helped reduce series resistance by providing a better ohmic contact for the base. Despite these improvements, the scalability of these devices remains limited due to their inherently low diffusion length [265].

### 4.2. Diamond Field Effect Transistors

Diamond MOSFETs are usually boron-doped p-channel MOSFETs (see Figure 10a) or transfer-doped devices. However, high activation energy for the dopants results in low conduction current. Due to diamond’s low-ionized impurity property, a unique method was developed to realize MOSFET devices. A slightly different structure than the conventional MOSFET was designed, called p-i-p metal-intrinsic FET (MISFET) [267]. Here, a trench is etched through the boron-doped layer, and gate dielectric and metal are deposited on the trench, as demonstrated in Figure 10b. The current passes around the gate through the channel by taking advantage of the better mobility and conductivity of the intrinsic diamond layer. However, this device is not suitable for high-power applications due to its space charge limited current mechanism [29,268]. Another design (called delta-doped FET) was introduced, where a thin layer of high-density boron dopant is sandwiched inside the intrinsic diamond layer [269]. The current conduction happens on both the intrinsic and the delta-doped layers, but there is a huge difference in the carrier mobilities between the delta-doped layer (1–10 cm^2^/Vs) and the intrinsic layer (3800 cm^2^/Vs) [270]. Research is going on to improve this condition [271]. Apart from that, fabricating that thin nanometer level of delta-doped layer is challenging. Figure 10c shows the schematic of the boron delta-doped FET. A low impurity concentration of intrinsic diamond has made it a good choice for creating a stable deep depletion region [272]. A deep depletion boron-doped diamond MOSFET (see Figure 10a) had a high breakdown field of 4 MV/cm [273]. Al_2_O_3_ was achieved for the gate dielectric for a normally ON lateral device. A normally OFF p-channel inversion type MOSFET has been reported as well (see Figure 10d) [274]. A phosphorous-doped n-type diamond <111> substrate body was utilized for the inversion mode MOSFET. As diamond MOSFET development hinders due to a lack of natural oxide, a high-quality gate dielectric Al_2_O_3_/O-terminated diamond interface was created by using OH termination through a wet annealing process. The device showed a maximum current density of 1.6 mA/mm. The extracted value of the channel mobility was reported to be 8 cm^2^/Vs owing to the interface trap density (6 × 10^12^ cm^−2^ eV^−1^). Later investigations showed that the low channel mobility is related to the roughness of the Al_2_O_3_/O-terminated diamond interface, which is proportional to the phosphorous dopant concentration [275].

The diamond metal semiconductor FET (MESFET) configuration has more advantages in high voltage applications as it does not have a gate dielectric oxide layer with a high density of interface states [256,274]. Boron-doped p-channel diamond MESFETs have exhibited breakdown over 2 kV with gate-to-drain length of 50 µm [276]. Among several Schottky gate metals, Pt showed a maximum current density of 0.06 mA/mm and 1.2 mA/mm at room temperature and high-temperature (300 °C) [277]. High resistance at the source and drain contacts with the diamond drift layer is a big concern for FET devices. This restricts the drain current immensely. A highly-doped p+ layer was grown below the metal contact at the source and the drain (shown in Figure 10e) to make good ohmic contact [278]. A three-fold reduction in ON resistance was reported, which resulted in a significant increment in the maximum current density from 0.06 mA/mm to 0.2 mA/mm at room temperature. Moreover, only recently has the first n-channel MESFET been created using the concept of a highly doped layer [279]. The highly doped phosphorous layer n+ (approximately 10^20^ cm^−3^) was sandwiched between the metal contacts and the n- (approximately 10^16^ cm^−3^) drift layer at the source and drain. However, the maximum current density was quite low (0.00093 µA/mm) at room temperature, which was elevated to 0.13 µA/mm at high-temperature (300 °C).

H-terminated or hydrogen surface channel FETs have been the best devices so far in terms of providing the best hole mobility and conductivity in the channel (schematic shown in Figure 10f) [280,281]. Different structures, such as MOSFET [280,282], MESFET [283,284], and MISFET [285], have been used. They usually have good high-frequency performance. However, there are some concerns about the long-term stability of the channel under high temperatures and harsh conditions. Using atomic layer deposition (ALD) of Al_2_O_3_ improves stability and reliability [282]. A high breakdown voltage of over 2 kV has been reported for the normally OFF MOSFET configuration [286]. Here, a partial C-O region was implemented to achieve the OFF operation. Another report showed a maximum current density of 1.3 A/mm. Both devices used Al_2_O_3_ as the passivation layer. Table 2 compares the parameter comparison of diamond high-power devices in comparison with the competing material systems’ devices.

## 5. Diamond-Based High-Frequency Devices

Surface-channel FETs are the only diamond devices supporting high-frequency operations. Typically, an H-terminated surface reduces the channel sheet resistance and gives rise to high hole mobility [280,281,288]. The mechanism of this process of surface charge accumulation is still uncertain. Different forms of surface acceptor medium help create a 2DHG channel under the H-terminated diamond surface [218]. In a MOSFET, the surface hole mobility is hampered by the dielectric/diamond surface interface traps [42,289]. Mobility enhancement was reported by using the monocrystalline h-BN (planner lattice structure) as the dielectric medium [42]. Some of the earliest reports on the high-frequency operation of MESFET H-FET can be found in [283,290]. The MISFET structure was chosen to limit the oxygen contamination on the diamond surface. A CaF_2_ layer was deposited as the insulating layer as it has low interface states with diamond [285]. For thin dielectric layers (below 30 nm), a high gate leakage current was a problem. Different diamond FET structures with various dielectric materials were studied, resulting in notably improved performance. Figure 11 shows various diamond H-FET designs showing superior high-frequency performance.

Al_2_O_3_ was used as the passivation layer, which blocks the gate leakage current because of its higher band offset value at the interface [292]. The Al thin film was oxidized to fabricate the Al_2_O_3_ layer, keeping the important ambient absorbate secured. Using this technique, a surface channel MOSFET was fabricated exhibiting 2.14 W/mm of power density at 1 GHz frequency with high current density (790 mA/mm) [293]. It also showed a cutoff frequency of 45 GHz. Another similar device showed a higher power density (3.8 W/mm) at 1 GHz (see Figure 12a). This device used 100 nm of the same dielectric material (Al_2_O_3_) deposited using atomic layer deposition at high-temperature (450 °C) [16,231]. The device had a cutoff frequency of 31 GHz and achieved the highest reported diamond hole drift velocity (1 × 10^7^ cm/s), close to the saturation velocity as shown in Figure 12b. A recently reported device achieved an output power density of 1.5 W/mm at 3.6 GHz (see Figure 12c) with high voltage operation supported by a thick ALD-deposited Al_2_O_3_ layer [291]_._ This is the highest reported output power density for diamond FET at frequencies above 2 GHz, as shown in Figure 12d. The self-aligned gate process [284] was used to fabricate the device. Moreover, several other H-FET MOSFET devices with self-oxidized Al_2_O_3_ showed notable performances, such as the power of 745 mW/mm at 2 GHz for a 0.8 µm gate length device [294], MOSFET with low temperature (90 °C) Al_2_O_3_ deposition (182 mW/mm at 10 GHz; gate length 0.1 µm) [295], and MISFET with self-aligned gate process (650 mW/mm at 10 GHz; gate length 0.35 µm) [296]. In summary, the technologies used in the devices showing good performances in terms of output power density are self-oxidized alumina, ALD Al_2_O_3_ devices with a larger gate-to-drain distance.

The maximum frequency reached by a diamond FET with a gate length of 100 nm was 120 GHz [14]. The device was fabricated using a polycrystalline, large grain, high quality diamond substrate with a (110) orientation. The state-of-the-art cutoff frequency achieved by diamond FET is 70 GHz in a MISFET configuration with a T-shape gate structure, and the length is 100 nm, as shown in Figure 13a [15]. Another high-performance diamond H-FET device showed a maximum functioning frequency of 63 GHz and had a gate length of 200 nm [297]. The maximum cutoff frequencies reached by the diamond RF FETs are comparable to the state-of-the-art AlGaN/GaN HEMT devices [17,298,299], as demonstrated in Figure 13b. On the other hand, Figure 13c shows that in terms of output power density, diamond FETs put behind most of the SiC FET, Si LDMOS, and GaAs FET devices. AlGaN/GaN HEMT devices are still leading in output power densities. Shorter channel lengths are used to increase the maximum operating frequency. However, it also reduces the output power density due to smaller device lengths, as shown in Figure 12d. Additionally, stability is an important aspect of high-performance RF devices. Numerous high-k dielectric material layers have been explored for stability and reliability performance tests. Al_2_O_3_ passivation with NO_2_ adsorption and ALD Al_2_O_3_ at high-temperature (450 °C) showed stable performance at 200 °C and 400 °C, respectively [280,300]. Moreover, MoO_3_ [301], V_2_O_5_ [302], and V_2_O_5_/Al_2_O_3_ [303] passivation layers resulted in no significant change in device performance for 17 days and a month-long period, respectively.

## 6. Plasma Wave in Diamond Terahertz Field Effect Transistors (Diamond TeraFETs)

The utilization of sub-terahertz (sub-THz) and terahertz (THz) frequency bands holds immense potential for a wide range of technologies, including advanced high-speed (6G) communications [1], VLSI testing [4,5], security systems, medical imaging [304], bio/chemical sensing [305], and THz spectroscopy. This creates a significant demand for compact, high-quality, room temperature-operating THz electronic sources and detectors. However, the development of THz electronic devices has been limited, as traditional electronic approaches struggle to achieve cutoff frequencies in the THz range by shortening gate length (see Figure 12b and Figure 13b). In recent years, there has been substantial research into terahertz plasmonic FETs [306,307,308,309]. These devices harness plasma waves, which are oscillations in charge density within 2D electron/hole gas systems in the device channels. These oscillations are akin to sound waves in musical instruments but are carried by charged particles that carry electromagnetic power. They can be employed for THz detection and emission. The FET devices operating in the THz range are called TeraFETs. Depending on channel length, mobility, and momentum relaxation time, plasmonic TeraFETs can operate in broadband or resonant detection modes. Material properties such as mobility and momentum relaxation time play a crucial role. Si-MOSFETs [310], GaAs/AlGaAs [311], and InGaAs/GaAs [312] HEMTs have been used for plasmonic TeraFET detectors. They provided room temperature broadband detection [313], and their noise-equivalent power at sub-THz and THz frequencies has been sufficiently low. More exotic materials, such as graphene [314,315] and diamond [19,20], have recently been explored due to their exceptional charge transport properties. Diamond, in particular, has potential advantages over other materials in terms of THz detection and emission, thanks to its long momentum relaxation time and high mobility [18]. Diamond-based TeraFETs have shown potential for both room temperature and cryogenic operation [316,317].

The concept of THz detection and emission using plasmonic oscillations within the 2DEG (2D electron gas) system was first introduced using the hydrodynamic model in 1993 [318,319]. This approach is simpler than the particle-based approach such as Monte-Carlo [320]. However, this method is applicable when the electrons or the holes in the channel can be treated as an electronic (or hole) fluid.

### 6.1. Hydrodynamic Model

In a high charge density channel, when the mean free path of electron-electron or hole-hole collision is much shorter than the channel length and the mean free path of electron/hole-impurity and lattice collision is longer than the channel length, the channel can be considered an electron/hole fluid. The hydrodynamic model is based on the Euler Equation (2) and the continuity Equation (3) [307,321].
(2)$$\frac{\partial n_{s}}{\partial t}+\nabla \cdot \left(n_{s}u\right)=0,$$
(3)$$\frac{\partial u}{\partial t}+u\nabla u+\frac{e}{m}\nabla U+\frac{u}{\tau }-v\nabla^{2}u=0,$$

Here, n_s_ is the charge density in the channel, u represents the drift velocity, U is the gate-to-channel potential, and v is the dynamic viscosity of the channel. The most important parameter is the momentum relaxation time of the charge carrier (τ=µm/e; µ = mobility, m = effective mass of the electron/hole). Following the gradual channel approximation, the charge density is calculated from CU = en_s_, where C is the per unit area capacitance. When THz radiation is impinged on an FET device, it excites a plasma oscillation in the channel’s shorted source-gate side, which travels between drain and source. Due to the nonlinear properties of the FET, this induced charge oscillation in the channel gets rectified and produces a source-to-drain DC voltage that is proportional to the incident signal’s amplitude. This is possible when an asymmetric boundary condition is applied between the drain (open boundary condition) and the source (short boundary condition). Under these boundary conditions, the system produces a DC voltage (∆U) as follows [321]:(4)$$\frac{∆U}{U_{0}}=\frac{1}{4}{\left(\frac{U_{a}}{U_{0}}\right)}^{2}f\left(\omega \right),$$

Here, U_a_ is the amplitude of the radiated THz signal, and U_0_ gate swing voltage. ∆U exhibits resonance with respect to the excited frequency (ω). Notably, it demonstrates its most pronounced response at the frequency of plasma oscillation and its odd harmonics. Also, if ωτ≫1, then the detection will be a resonant detection, but for ωτ≪1, it will be broadband detection. ω = πn/(2L) [307].

THz emission requires different boundary conditions with a constant DC current. The Dyakonov-Shur instability occurs when the drift velocity is high enough to overcome the damping scattering and the viscosity damping [318]. The wave travels from source to drain and gets reflected with the gain due to the asymmetric boundary conditions. In every turn, the amplitude of the plasma wave increases, and over time, it produces a full form standing wave in the channel. The earliest report derived that when the drift velocity is less than the plasma wave velocity (s) in the channel, the plasma oscillation frequency follows the following equation [318].
(5)$$\omega^{\prime }=\frac{\left|s^{2}-u^{2}\right|}{2Ls}\pi n,$$

Here L is the channel length, and n is an odd integer number. Once this plasma oscillation sustains itself in the charge density of the channel, it creates mirror charge density oscillations in the gate metal and produces EM radiation at the plasma frequency.

### 6.2. Diamond Properties for THz Detection and Emission

Diamond’s exceptional physical properties make it very suitable for THz plasmonic devices. When it comes to electron and hole mobilities in diamond, the highest reported values obtained through time-resolved cyclotron resonance are 7300 cm^2^/Vs and 5300 cm^2^/Vs, respectively [21]. Additionally, diamond exhibits substantial surface-heavy hole effective mass values, typically ranging from 0.66 to 2.12 [197,322]. Table 3 lists the important parameters for THz detection and emission for different materials, including diamond. Diamond’s large optical phonon energy serves to minimize optical phonon scattering, resulting in a significantly extended momentum relaxation time, particularly limited by optical phonon scattering. This property positions p-diamond as a highly promising candidate for sub-THz and THz plasmonic devices. Notably, the quality factor (Q = ωτ) of plasmonic resonance in p-diamond, can potentially exceed unity at frequencies as low as 300 GHz [18].

Recently, p-diamond has emerged as a contender for plasmonic THz applications, drawing attention to its substantial potential for THz and IR plasmonic detection [18]. One crucial parameter governing the plasma response in p-diamond is the hole momentum relaxation time (τ), which can vary significantly, ranging from 0.013 to 6.4 ps at room temperature. Figure 14a shows that p-diamond crosses the unity value of the quality factor in the sub-THz frequency range at room temperature, which is required for resonant THz detection and emission. Also, p-diamond has the longest reported hole momentum relaxation time among other competitor materials, which helps in reaching the high-quality factor as shown in Figure 14b,c. The minimum mobility required for p-diamond to achieve resonant operation is much lower than that of Si, GaN, and InGaAs, as shown in Figure 14d [19]. This makes p-diamond a superior material of choice for TeraFET applications.

Moreover, p-diamond TeraFETs can support better resonant operation (comparing quality factor values) than Si, GaN, and InGaAs at cryogenic (77 K) and room (300 K) temperatures [317]. Si can achieve plasmonic oscillation at cryogenic temperature with 22 nm and 65 nm of channel lengths only if the highest reported mobility (1450 cm^2^/Vs) is ensured. On the other hand, a p-diamond FET can function as a THz plasmonic resonator even with 130 nm of channel length at room temperature. All these studies predict that diamond could be a promising candidate for TeraFET applications.

### 6.3. Terahertz Detection and Emission by Diamond TeraFET

Various analytical and numerical studies have been conducted to analyze the THz detection and emission process by diamond TeraFET using hydrodynamic modeling [19,20,324,325,326]. Figure 15a,b shows the diagrams of the detection and emission processes using diamond TeraFETs. Numerical simulations were reported on the resonant response of sub-THz and THz radiation in p [19,20,327] and n-diamond [324,326] TeraFET at cryogenic and room temperature, as predicted in the earlier analysis of the diamond’s material properties. P-diamond TeraFETs with over 100 nm channel length should be able to achieve a resonant response even at room temperature.

Among different material systems such as InGaAs, Si, and GaN, the DC responsivity values for diamond TeraFET were reported to be the highest, as shown in Figure 16a. Also, p-diamond has the lowest resonance frequency for the same biasing condition and the same channel length. This demonstrates that it will be easier for p-diamond to detect the 300 GHz resonance response, which is promising for implementing 300 GHz communication links. Figure 16b shows the DC response of p-diamond at cryogenic (77 K) and room temperature (300 K), where the channel length was 130 nm. Moreover, the gate biasing dependency of the resonance frequency in both p- and n-diamond can help realize a tunable THz detector.

Ultrashort pulse detection analysis has been demonstrated numerically in diamond TeraFET [328]. To boost the DC response, a variable gate structure was investigated. p-diamond showed higher responsibilities among other material systems [329].

Compact modeling of diamond TeraFETs uses transmission line analysis for p-implemented SPICE and ADS modeling [317]. Apart from that, the operation of a TeraFET THz spectrometer has been reported in [330,331,332]. SPICE modeling has been used for the analysis, which demonstrated that p-diamond supported sub-THz operation in the 200–600 GHz frequency range for a channel length of 250 nm.

Recent study have reported the numerical analysis of sub-THz and THz emission from p-diamond TeraFET by implementing the Dyakonov-Shur instability in the channel [327]. P-diamond showed plasma wave oscillation in the frequency range from 200–400 GHz for channel lengths of 80 to 120 nm at room temperature, as shown in Figure 16c,d. This makes p-diamond an interesting room temperature compact THz source for the 300 GHz communication technology.

However, there has been no report on the experimental demonstration of these features in diamonds yet. Though these reports show great promise for the diamond-based TeraFET, but fabricating a diamond-based FET with high mobility and making good ohmic contacts at the source and drain will be challenging. New designs and techniques (rachet [333], perforated channel [334], and plasmonic crystal [335]) have been proposed to achieve the highest predicted performances.

## 7. Conclusions

In summary, we reviewed the state-of-the-art diamond high-power and frequency devices in this paper. Diamond promises superlative performance in the field of high-power and high-frequency applications due to its exceptional material properties. However, the lack of efficient dopant atoms held down the progress of producing high quality diamond devices. Though many of the bottlenecks have been resolved over the years and high-quality diamond substrates are being manufactured; substrate size, quality of the lattice, and n-type doping remain some of the issues that need to be resolved. Over the years, various device structures, including Schottky diodes, p-i-n diodes, p-n diodes, and FETs, have been developed using diamond, displaying competitive performance in high-power applications compared to other materials. However, there is room for improvement, especially in the development of diamond bipolar devices to achieve top-tier performance in high-power applications. When it comes to state-of-the-art diamond RF devices, they exhibit impressive performance in terms of cutoff and maximum frequency, as well as output power density, surpassing many other material systems. Yet, they still fall short of the performance achieved by AlGaN/GaN HEMT devices. Fortunately, diamond’s long carrier momentum relaxation time has opened up new possibilities in the sub-THz and THz frequency ranges, leveraging plasma waves in TeraFETs and moving beyond traditional electronics approaches. Diamond has the potential to enable room temperature operation in the mm-Wave frequencies, paving the way for advanced THz applications, including 300 GHz communication.

## Figures and Tables

**Figure 1 nanomaterials-14-00460-f001:**
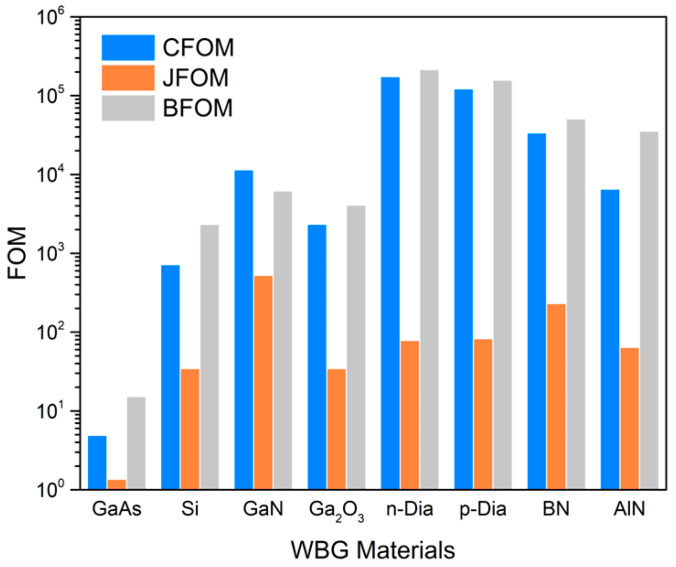
Baliga [22], Johnson [23], and combined [24] of semiconductor materials (relative to silicon).

**Figure 2 nanomaterials-14-00460-f002:**
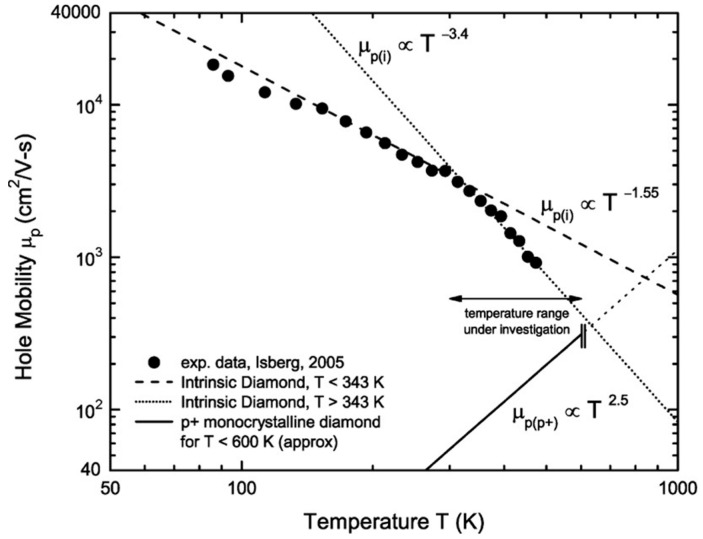
Mobility of p-diamond depending on temperature [43].

**Figure 3 nanomaterials-14-00460-f003:**
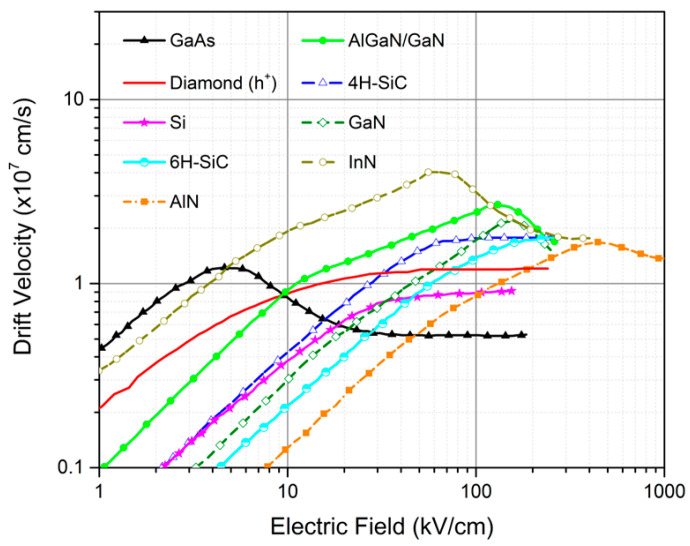
Charge drift velocity vs. electric field of different wide band gap semiconductors [52,53,54].

**Figure 4 nanomaterials-14-00460-f004:**
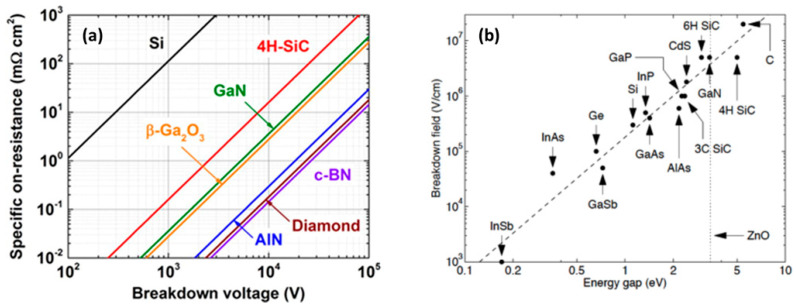
(**a**) Specific ON resistance vs. breakdown voltage of various semiconductor materials at ambient temperature [27,56], (**b**) breakdown field vs. energy band gap of different semiconductor materials [57].

**Figure 5 nanomaterials-14-00460-f005:**
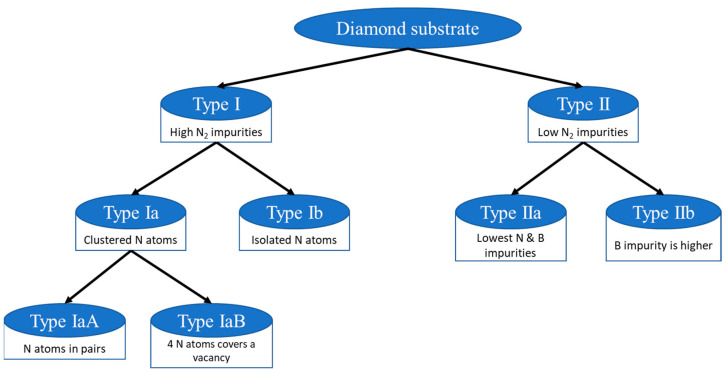
A diagram of the diamond substrate’s classification.

**Figure 6 nanomaterials-14-00460-f006:**
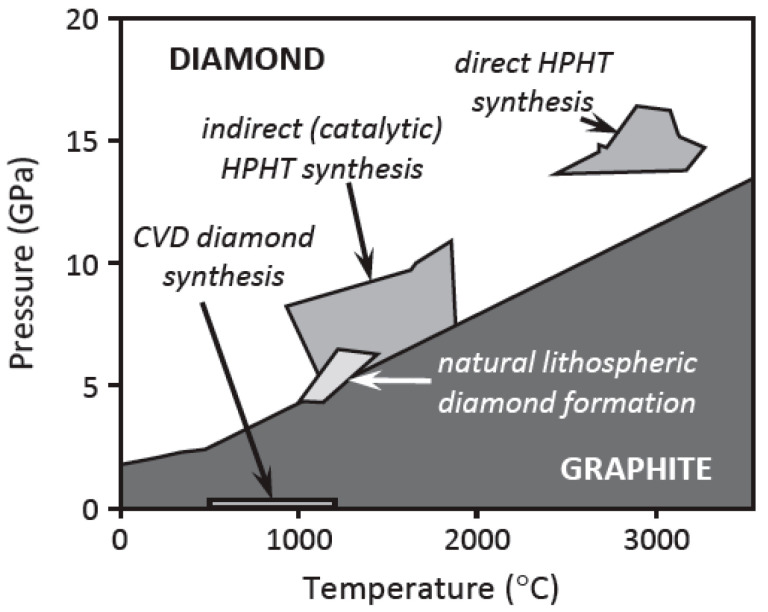
Phase diagram of carbon [64].

**Figure 7 nanomaterials-14-00460-f007:**
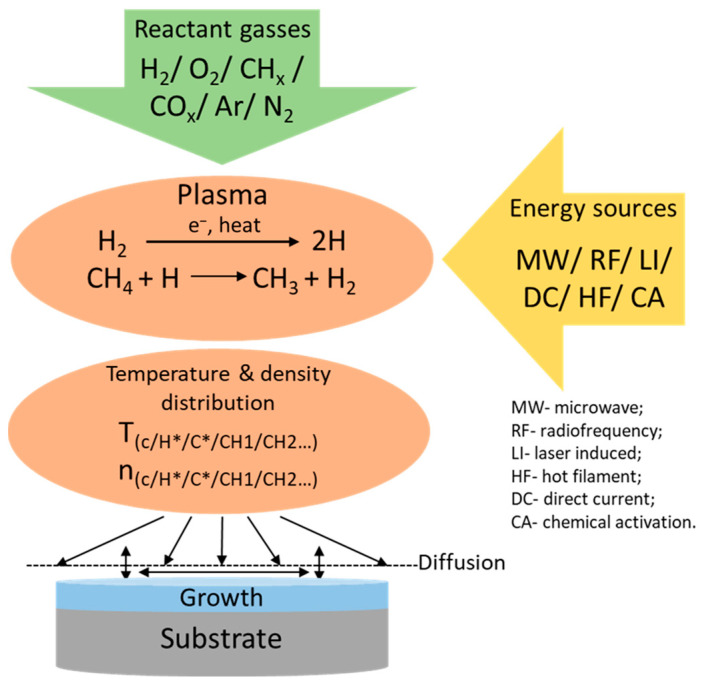
Process flow diagram for diamond CVD growth [80].

**Figure 8 nanomaterials-14-00460-f008:**
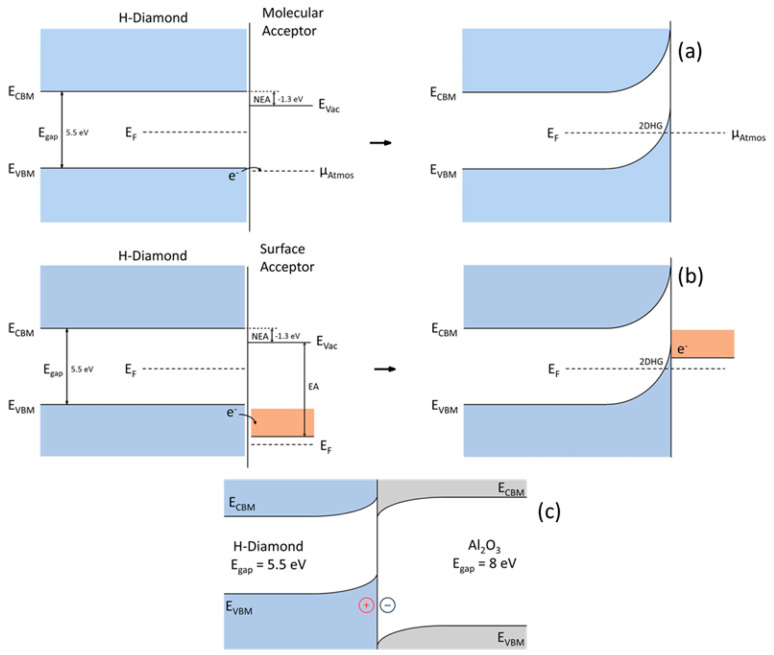
Diagram of the 2DHG charge accumulation process when the surface acceptor is (**a**) ambient air, (**b**) with high electron affinity, and (**c**) Al_2_O_3_ [218].

**Figure 9 nanomaterials-14-00460-f009:**
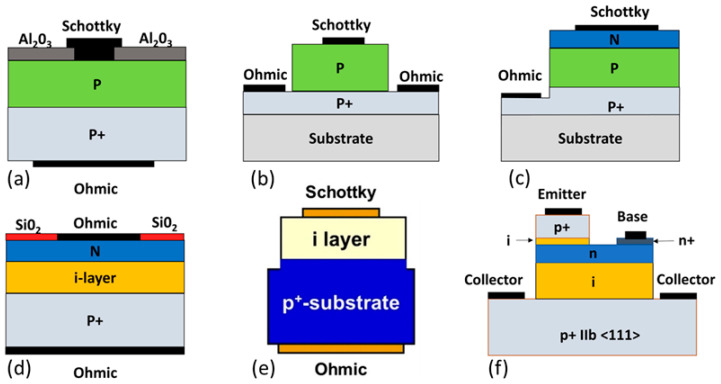
Structures of different diamond junction power devices (**a**) vertical SBD, (**b**) pseudo-vertical SBD, (**c**) Schottky PND, (**d**) pin diode, (**e**) metal-i-P diode, and (**f**) BJT [28,256].

**Figure 10 nanomaterials-14-00460-f010:**
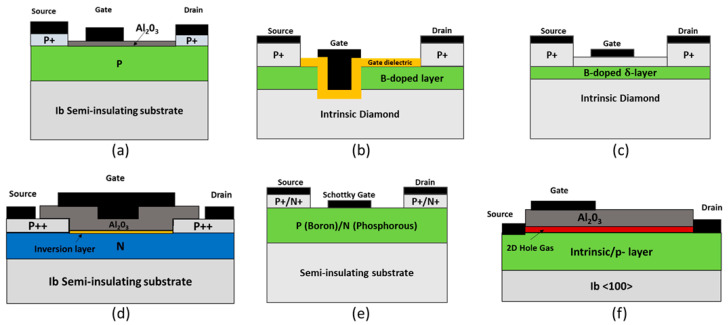
Structures of different diamond FET configurations: (**a**) deep depletion MOSFET, (**b**) p-i-p metal-intrinsic FET (MISFET), (**c**) delta-doped FET, (**d**) inversion mode MOSFET, (**e**) p/n-channel MESFET, and (**f**) H-terminated FET (H-FET) [256].

**Figure 11 nanomaterials-14-00460-f011:**
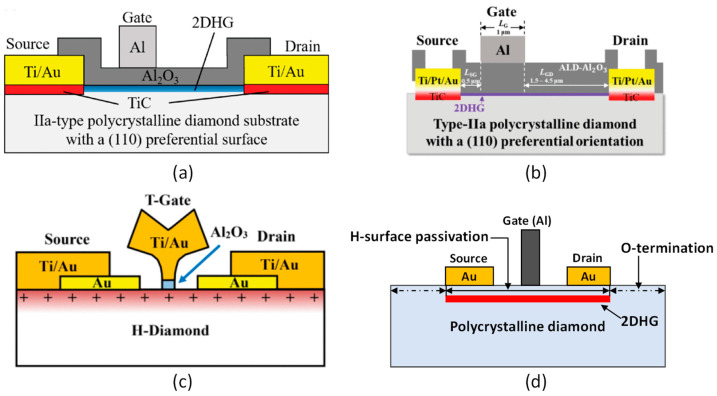
Diagrams of different diamond H-FET RF high-frequency devices: (**a**) MOSFET—3.8 W/mm at 1 GHz [16], (**b**) MOSFET with thick Al_2_O_3_ layer—1.5 W/mm at 3.6 GHz [291], (**c**) MISFET—70 GHz cutoff frequency [15], and (**d**) MESFET—120 GHz maximum frequency [14].

**Figure 12 nanomaterials-14-00460-f012:**
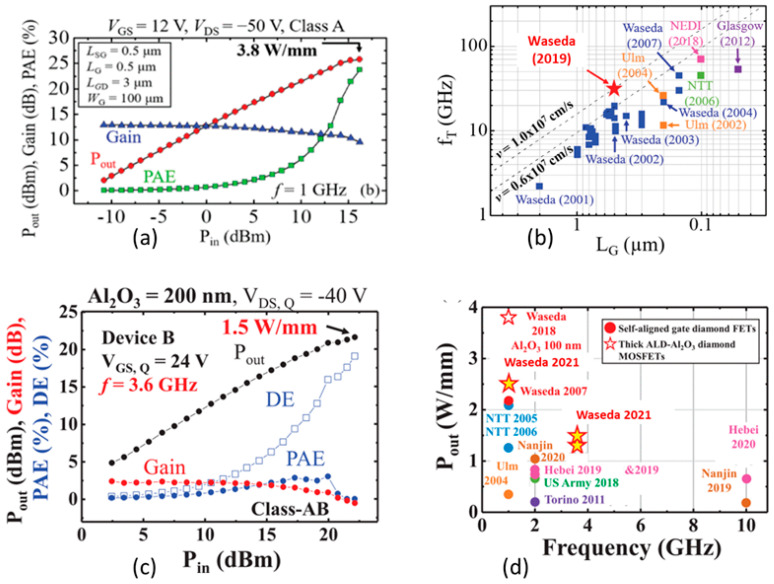
(**a**) Large signal performance for MOSFET showing highest output power 3.8 W/mm at 1 GHz [16], (**b**) Cutoff frequency vs. gate length for diamond FET devices (dotted lines indicating drift velocity) [16], (**c**) Large signal performance for MOSFET with thick Al_2_O_3_ layer showing highest output power 1.5 W/mm at 3.6 GHz [291], and (**d**) Output power density vs. frequency for different diamond devices (“star symbol at 3.6 GHz”: highest reported output power from diamond FET over 2 GHz) [291].

**Figure 13 nanomaterials-14-00460-f013:**
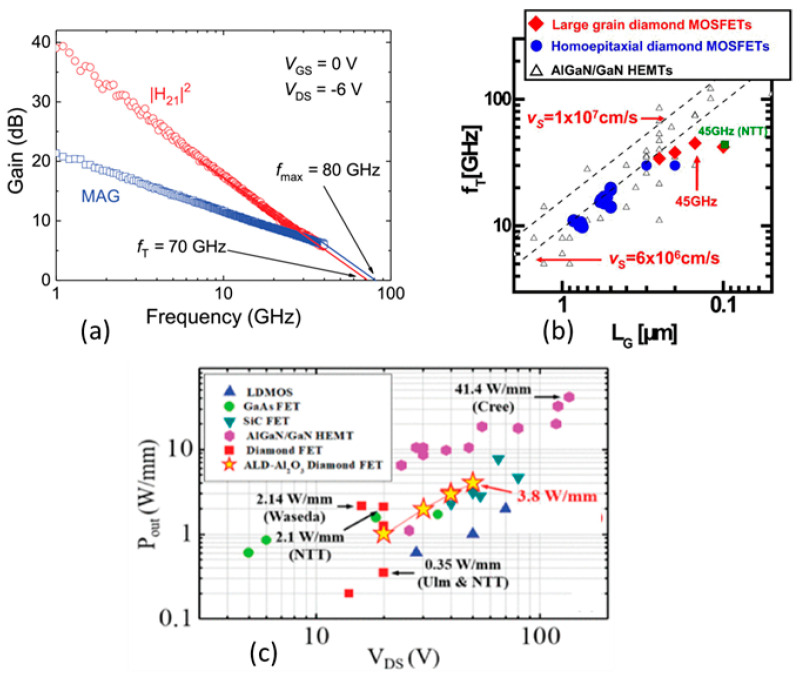
(**a**) Small signal analysis for T-gate MISFET showing the highest cutoff frequency at 70 GHz [15], (**b**) Cutoff frequency vs. gate length for diamond FET and AlGaN/GaN HEMT devices [293], (**c**) Output power density vs. drain bias for RF devices with different material systems, including diamond [16].

**Figure 14 nanomaterials-14-00460-f014:**
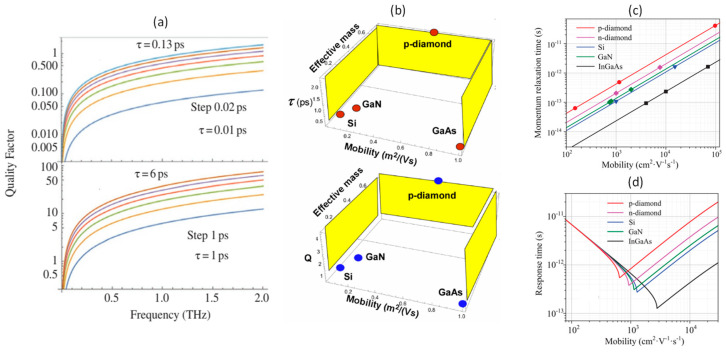
(**a**) p-diamond quality factor (Q) vs. frequency for room temperature momentum relaxation times [18,323], (**b**) momentum relaxation time and quality factor (Q) of different materials as a function of mobility and effective mass at 300 GHz [324], (**c**) momentum relaxation time of different materials for measured values of mobility [19], and (**d**) ultimate response times for different materials vs. their carrier mobilities [19].

**Figure 15 nanomaterials-14-00460-f015:**
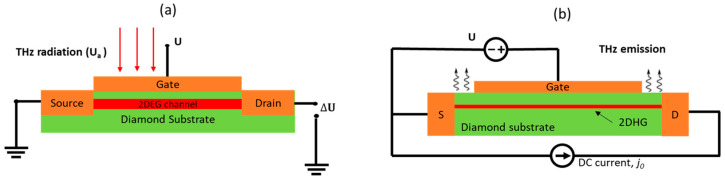
Schematic of THz (**a**) detection and (**b**) emission using diamond TeraFET.

**Figure 16 nanomaterials-14-00460-f016:**
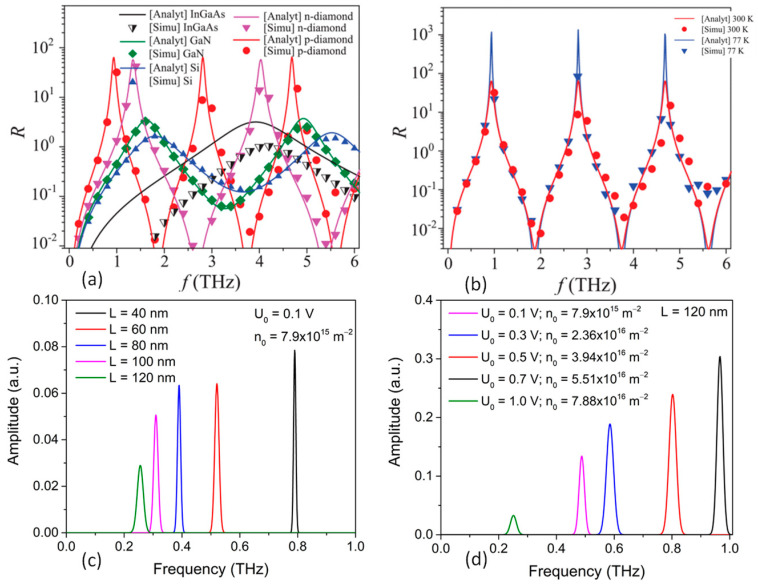
Normalized DC response of THz detection for (**a**) different materials [19], (**b**) p-diamond at 77 K and 300 K [19], (**c**,**d**) sub-THz and THz emission from p-diamond TeraFET ($n_{0}$ is charge density and $U_{0}$ is the gate swing voltage) [327].

**Table 2 nanomaterials-14-00460-t002:** Comparison of high-power device performance made from different material systems [27,28,256,287].

Material	Devices	Breakdown Voltage	Maximum Current	Gate to Drain Length	Breakdown Field
AlGaN/GaN	FET	200–1400 V	300 mA/mm	4–20 µm	1 MV/cm
AlGaN/AlGaN	FET	500–1700 V	200 mA/mm	1–10 µm	1.7 MV/cm
Ga_2_O_3_	MOSFET	400 V	60 mA/mm	8 µm	0.5 MV/cm
SiC	FET	1600 V	90 mA/mm	20 µm	0.8 MV/cm
	Lateral SBD	Over 10,000 V	18 A/cm^2^	300 µm	-
Diamond	Vertical SBD	Over 1800 V	Over 100 A/cm^2^	-	-
	Pseudo-vertical SBD	Over 1600 V	4.5 kA/cm^2^	-	7.7 MV/cm
	PiN Diode	Over 11,000 V	Below 10 A/cm^2^	-	-
	Schottky pn Diode	Below 55 V	Over 60 kA/cm^2^	-	-
	BJT	Below 100 V	approximately µA range	-	-
	MOSFET	Over 200 V	Below 1 mA/mm	-	4 MV/cm
	MESFET	approximately 3000 kV	approximately 2 mA/mm	30 µm	approximately 2 MV/cm
	H-FET	Over 2000 V	1.3 A/mm	21 µm	-

**Table 3 nanomaterials-14-00460-t003:** Diamond, GaN, InGaAs, and Si properties for TeraFET applications [19].

Material	Effective Mass	Mobility (300 K) (cm^2^/Vs)
p-diamond	0.74	5300
n-diamond	0.36	7300
GaN	0.24	2000
InGaAs	0.041	12,000
Si	0.19	1450

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
