# Peer review of "Diamond for High-Power, High-Frequency, and Terahertz Plasma Wave Electronics"

_nanomaterials, 2024, doi:10.3390/nano14050460_

Round 1

Reviewer 1 Report

Comments and Suggestions for Authors

The paper, titled "Diamond Technology: Unveiling the Potential for High-Power and High-Frequency Semiconductor Devices," provides an insightful exploration into the promising realm of diamond as a candidate material for high-power and high-temperature semiconductor devices. The manuscript outlines the key attributes of diamond, such as its high thermal conductivity, high breakdown field, and superior radiation hardness compared to silicon, making it an appealing choice for challenging applications.

One of the significant strengths highlighted in the manuscript is diamond's exceptional electron and hole momentum relaxation times, facilitating the development of compact terahertz (THz) and sub-THz plasmonic sources and detectors. The ability to operate at both room and elevated temperatures is a notable advantage, indicating the potential for diamond-based devices to thrive in extreme environments.

The manuscript further emphasizes the potential of diamond TeraFETs (Terahertz Field-Effect Transistors) in the 240-600 GHz atmospheric window, showcasing a plasmonic resonance quality factor larger than unity. This specific quality factor could position diamond TeraFETs as viable candidates for 6G communications applications, underscoring their relevance in the ever-evolving landscape of high-frequency technologies.

The comprehensive review not only highlights the potential advantages of diamond technology but also acknowledges the challenges that need to be addressed. By doing so, the paper provides a balanced perspective on the feasibility and limitations of incorporating diamond in semiconductor devices.

The clarity of the manuscript and its well-structured content make it an engaging read for both experts in the field and those seeking to understand the advancements in semiconductor materials. Overall, this paper contributes valuable insights into the potential role of diamond in augmenting silicon for high-power and high-frequency compact devices, with a particular focus on extreme environments and high-frequency applications.

Author Response

Reviewer #1: The paper, titled "Diamond Technology: Unveiling the Potential for High-Power and High-Frequency Semiconductor Devices," provides an insightful exploration into the promising realm of diamond as a candidate material for high-power and high-temperature semiconductor devices. The manuscript outlines the key attributes of diamond, such as its high thermal conductivity, high breakdown field, and superior radiation hardness compared to silicon, making it an appealing choice for challenging applications.

One of the significant strengths highlighted in the manuscript is diamond's exceptional electron and hole momentum relaxation times, facilitating the development of compact terahertz (THz) and sub-THz plasmonic sources and detectors. The ability to operate at both room and elevated temperatures is a notable advantage, indicating the potential for diamond-based devices to thrive in extreme environments.

The manuscript further emphasizes the potential of diamond TeraFETs (Terahertz Field-Effect Transistors) in the 240-600 GHz atmospheric window, showcasing a plasmonic resonance quality factor larger than unity. This specific quality factor could position diamond TeraFETs as viable candidates for 6G communications applications, underscoring their relevance in the ever-evolving landscape of high-frequency technologies.

The comprehensive review not only highlights the potential advantages of diamond technology but also acknowledges the challenges that need to be addressed. By doing so, the paper provides a balanced perspective on the feasibility and limitations of incorporating diamond in semiconductor devices.

The clarity of the manuscript and its well-structured content make it an engaging read for both experts in the field and those seeking to understand the advancements in semiconductor materials. Overall, this paper contributes valuable insights into the potential role of diamond in augmenting silicon for high-power and high-frequency compact devices, with a particular focus on extreme environments and high-frequency applications.

Author response:  We sincerely appreciate your inspiring comments and feedback on the manuscript.

Reviewer 2 Report

Comments and Suggestions for Authors

Diamond is an extensively utilized material in an extensive range of photonic and quantum potential applications involving high power and high temperature semiconductor devices or devices requiring radiation hardness. The plasmonic resonance quality factor in diamond emerging at the 240-600 GHz atmospheric window which could make them viable for 6G communications applications. The electronic material properties of diamond are mentioned, the material quality and growth techniques are analyzed while the diamond-based high power electronic devices are described. Plasma wave in diamond terahertz field effect transistors (diamond TeraFETs) are also proposed as applicable for the efficient operation of communication systems. 

It is an extensive and informative review on the use of diamond as a potential material in devices and electronic components covering a broad range of applications involving THz waves (6G communications, renewable energy transport and distribution, power electronics etc). The paper can be published at Nanomaterials as long as the potential to use diamond towards quantum applications [1-3] is stressed more in a revised version. Such a modification will complete the useful review at hand.

[1] Diamond quantum thermometry: from foundations to applications, Nanotechnology, 2021.

[2] Quantum Fabry-Perot Resonator: Extreme Angular Selectivity in Matter-Wave Tunneling, Physical Review Applied, 2019.

[3] Diamond quantum sensors: from physics to applications on condensed matter research, Functional Diamond, 2021.

Author Response

Reviewer #2: Diamond is an extensively utilized material in an extensive range of photonic and quantum potential applications involving high power and high temperature semiconductor devices or devices requiring radiation hardness. The plasmonic resonance quality factor in diamond emerging at the 240-600 GHz atmospheric window which could make them viable for 6G communications applications. The electronic material properties of diamond are mentioned, the material quality and growth techniques are analyzed while the diamond-based high power electronic devices are described. Plasma wave in diamond terahertz field effect transistors (diamond TeraFETs) are also proposed as applicable for the efficient operation of communication systems.

Reviewer #2, Comment #1: It is an extensive and informative review on the use of diamond as a potential material in devices and electronic components covering a broad range of applications involving THz waves (6G communications, renewable energy transport and distribution, power electronics, etc). The paper can be published at Nanomaterials as long as the potential to use diamond towards quantum applications [1-3] is stressed more in a revised version. Such a modification will complete the useful review at hand.

[1] Diamond quantum thermometry: from foundations to applications, Nanotechnology, 2021.

[2] Quantum Fabry-Perot Resonator: Extreme Angular Selectivity in Matter-Wave Tunneling, Physical Review Applied, 2019.

[3] Diamond quantum sensors: from physics to applications on condensed matter research, Functional Diamond, 2021.

Author response: We sincerely appreciate your valuable comments and suggestions. However, the current scope of our manuscript is focused on high power, high frequency, and terahertz diamond electronic devices. We will consider the suggested topic for a different paper in the future.

Reviewer 3 Report

Comments and Suggestions for Authors

The manuscript entitled “Diamond for High Power, High Frequency, and Terahertz Plasma Wave Electronics” by Muhammad Mahmudul Hasan, Chunlei Wang, Nezih Pala, and Michael Shur is of a review nature and does not contain new research results. Nevertheless, it is a valuable publication containing a huge amount of information about the current scientific achievements in research on diamond as a material with great potential for application in electronics. The authors collected and presented the results of research on the semiconductor properties of diamond doped with various elements and compared them with the results of other widely researched and used materials of this type. The authors pointed out that due to the wide energy bandgap, diamond is characterized by a very large electric breakdown field and exceptionally high thermal conductivity. Moreover, among materials with a wide energy bandgap, diamond is characterized by the highest mobility of charge carriers. The authors presented the classification of diamond substrates according to their quality and discussed in detail their production technologies, high pressure high temperature (HPHT) and chemical vapor deposition (CVD). They also discussed, in the context of other semiconductor materials, the properties of p-type doped and n-type doped diamond. The authors then presented a number of examples of diamond-based high power electronic devices, in particular Schottky diodes and various types of field effect transistors. Finally, the authors pointed out the potential possibility of using diamond-based semiconductor materials to build transistors that can act as detectors and emitters of terahertz radiation (THzFET). In this connection, they recalled the hydrodynamic model of plasma oscillations in a semiconductor developed by one of the authors in the 1990s. The manuscript contains a very extensive literature list containing 341 items, which, like the entire publication, can be very helpful for researchers dealing with diamond in terms of applications in electronics, and may also be interesting and inspiring for scientists who do not strictly deal with this topic.

The article presents a high substantive level. Therefore, I recommend the manuscript for publication in Nanomaterials (MDPI).

Please explain the “theta” symbol in formula (1) and the abbreviation "BJT" (line 467) in the text.

Author Response

Reviewer #3: The manuscript entitled “Diamond for High Power, High Frequency, and Terahertz Plasma Wave Electronics” by Muhammad Mahmudul Hasan, Chunlei Wang, Nezih Pala, and Michael Shur is of a review nature and does not contain new research results. Nevertheless, it is a valuable publication containing a huge amount of information about the current scientific achievements in research on diamond as a material with great potential for application in electronics. The authors collected and presented the results of research on the semiconductor properties of diamond doped with various elements and compared them with the results of other widely researched and used materials of this type. The authors pointed out that due to the wide energy bandgap, diamond is characterized by a very large electric breakdown field and exceptionally high thermal conductivity. Moreover, among materials with a wide energy bandgap, diamond is characterized by the highest mobility of charge carriers. The authors presented the classification of diamond substrates according to their quality and discussed in detail their production technologies, high pressure high temperature (HPHT) and chemical vapor deposition (CVD). They also discussed, in the context of other semiconductor materials, the properties of p-type doped and n-type doped diamond. The authors then presented a number of examples of diamond-based high power electronic devices, in particular Schottky diodes and various types of field effect transistors. Finally, the authors pointed out the potential possibility of using diamond-based semiconductor materials to build transistors that can act as detectors and emitters of terahertz radiation (THzFET). In this connection, they recalled the hydrodynamic model of plasma oscillations in a semiconductor developed by one of the authors in the 1990s. The manuscript contains a very extensive literature list containing 341 items, which, like the entire publication, can be very helpful for researchers dealing with diamond in terms of applications in electronics, and may also be interesting and inspiring for scientists who do not strictly deal with this topic.

The article presents a high substantive level. Therefore, I recommend the manuscript for publication in Nanomaterials (MDPI).

Reviewer #3, Comment #1: Please explain the “theta” symbol in formula (1) and the abbreviation "BJT" (line 467) in the text.

Author response: We sincerely appreciate your inspiring comments and feedback on the manuscript. The “theta” symbol in formula (1) was a typing mistake.

Author action: We removed the “theta” symbol from formula (1) and added the spelled out the abbreviation of “BJT”.